American Society for Microbiology

# *Candida albicans* Reactive Oxygen Species (ROS)-Dependent Lethality and ROS-Independent Hyphal and Biofilm Inhibition by Eugenol and Citral

Zinnat Shahina,[a] Easter Ndlovu,[a] Omkaar Persaud,[a] Taranum Sultana,[a] Tanya E. S. Dahms[a]

[a]Department of Chemistry and Biochemistry, University of Regina, Regina, Saskatchewan, Canada

**ABSTRACT** *Candida albicans* is part of the normal human flora but is most frequently isolated as the causative opportunistic pathogen of candidiasis. Plant-based essential oils and their components have been extensively studied as antimicrobials, but their antimicrobial impacts are poorly understood. Phenylpropenoids and monoterpenes, for example, eugenol from clove and citral from lemon grass, are potent antifungals against a wide range of pathogens. We report the cellular response of *C. albicans* to eugenol and citral, alone and combined, using biochemical and microscopic assays. The MICs of eugenol and citral were 1,000 and 256 $\mu$g/mL, respectively, with the two exhibiting additive effects based on a fractional inhibitory concentration index of 0.83 $\pm$ 0.14. High concentrations of eugenol caused membrane damage, oxidative stress, vacuole segregation, microtubule dysfunction and cell cycle arrest at the $G_1/S$ phase, and while citral had similar impacts, they were reactive oxygen species (ROS) independent. At sublethal concentrations (1/2 to 1/4 MIC), both oils disrupted microtubules and hyphal and biofilm formation in an ROS-independent manner. While both compounds disrupt the cell membrane, eugenol had a greater impact on membrane dysfunction. This study shows that eugenol and citral can induce vacuole and microtubule dysfunction, along with the inhibition of hyphal and biofilm formation.

**IMPORTANCE** *Candida albicans* is a normal resident on and in the human body that can cause relatively benign infections. However, when our immune system is severely compromised (e.g., cancer chemotherapy patients) or underdeveloped (e.g., newborns), this fungus can become a deadly pathogen, infecting the bloodstream and organs. Since there are only a few effective antifungal agents that can be used to combat fungal infections, these fungi have been exposed to them over and over again, allowing the fungi to develop resistance. Instead of developing antifungal agents that kill the fungi, some of which have undesirable side effects on the human host, researchers have proposed to target the fungal traits that make the fungus more virulent. Here, we show how two components of plant-based essential oils, eugenol and citral, are effective inhibitors of *C. albicans* virulence traits.

**KEYWORDS** *Candida albicans*, essential oils, cell death, virulence inhibition, citral, eugenol

Address correspondence to Tanya E. S. Dahms, tanya.dahms@uregina.ca.

The authors declare no conflict of interest.

**C**andida albicans, the most common cause of candidiasis, is an opportunistic pathogen that can cause thrush in the mouth or throat (1), vaginal yeast infections (2), or systemic candidiasis (3). In severely immunocompromised individuals, like those with AIDS, undergoing chemotherapy or organ transplant recipients, candidiasis is life-threatening, with high morbidity and mortality (4–6). The ability of *C. albicans* to gain access to deep tissues for systemic infection is invariably associated with adhesion, hyphal and biofilm formation (7, 8). Biofilms can form following adhesion to host tissues or to medical indwelling devices such as cardiovascular catheters, endotracheal

tubes, and cerebrospinal fluid shunts. Unfortunately, *C. albicans* is rapidly gaining resistance to a limited number of commonly used antifungal agents. Antifungal resistance makes the treatment of fungal infections often ineffective, generating an urgent need to discover novel antifungals for treatment or prophylaxis. In this context, plant-based essential oils (EOs) and their components (EOCs) are gaining popularity based on their strong antimicrobial and antibiofilm activities (9).

Eugenol (4-allyl-2-methoxyphenol) is a major constituent of clove essential oil most commonly derived from the buds and leaves of the aromatic plant *Eugenia caryophyllata* (or *Syzygium aromaticum* L.). Eugenol has many applications based on its large spectrum of biological activity (10–13). Studies examining the mechanism of eugenol antifungal activity underscore the importance of the phenolic group (14, 15), for which its hydrogen bonding capability and acidity are proposed to contribute to yeast (*Saccharomyces cerevisiae*) cell membrane disruption (16) by altering fluidity and permeability (17). Membrane compromise leads to cytoplasmic leakage (18, 19) and reactive oxygen species (ROS) accumulation (20), ultimately interfering with *C. albicans* adhesion and biofilm formation and viability (21–23). Citral (3,7-dimethyl-2,6-octadienal), also having antimicrobial properties, is an aliphatic aldehyde and the most abundant (65 to 85%) component of lemongrass (*Cymbopogon citratus*) essential oil (24). Citral causes cell membrane dysfunction, inhibition of respiratory enzymes, dissipation of the proton-motive force (25), leakage of cellular constituents (26, 27), and cell death, but can also potently inhibit *Candida* hyphal, mycelial, and biofilm growth (27, 28). Interestingly, both eugenol and citral arrest the *C. albicans* cell cycle, which has been attributed to membrane defects (29, 30). Eugenol and citral have been studied in combination with conventional antifungal agents (31–34), but their combined effect on *Candida* spp. remains unexplored (35). The synergistic effects of these EOCs against *Shigella flexneri*, *Aspergillus niger*, and *Penicillium roqueforti* (36–38) raise interest in their impact on *C. albicans*.

*C. albicans* lethality by eugenol and citral has been associated with cell membrane perturbation (17, 25, 26) and oxidative stress (20), but much less is known about their impact on virulence. Although the inhibition of candidal adhesion, biofilm formation, and mature biofilm viability by eugenol (21–23, 27, 28) and citral (26, 39, 40) has been documented, a clear understanding of the associated mechanisms is lacking. The anti-candidal and antivirulence activities of eugenol, citral, and their combination examined by epifluorescence, laser scanning confocal microscopy (LSCM), atomic force microscopy (AFM), and biochemical assays reveals ROS-dependent *C. albicans* cell membrane defects, cell cycle arrest, and cell death. However, the disruption of vacuoles, microtubules (MTs), and hyphal formation, the latter being a key *C. albicans* virulence factor which enables biofilm formation, is ROS independent.

## RESULTS

**Eugenol and citral are additive against *C. albicans*.** *C. albicans* RSY150 growth was effectively inhibited by 1,000 $\mu$g/mL eugenol and 256 $\mu$g/mL citral (Fig. 1a and b), and the two oils had an additive effect (fractional inhibitory concentration index [FICI] = 0.83 $\pm$ 0.14), as shown by the partial concave curve of the isobologram (Fig. 1c). In the presence of citral, the MIC of eugenol (454.5 $\mu$g/mL) was 2-fold lower than that of eugenol alone (1,000 $\mu$g/mL) (see Fig. S1 in the supplemental material).

**Eugenol and citral alter yeast morphology.** To address the impact of eugenol and citral on morphology, *C. albicans* RSY150 were challenged and imaged. Under standard conditions, *C. albicans* had regular yeast morphology with budding yeast cells, but following exposure to eugenol and citral, there were small tube-like filaments known as pseudohyphae appearing as chains or branched (Fig. 2a). Eugenol treatment led to a significant ($P < 0.0001$ and $P < 0.001$ for MIC and 1/2 MIC, respectively) increase in the number of chain forming-pseudohyphae (Fig. 2d), whereas citral mostly generated a significant ($P < 0.001$ and $P < 0.01$ for MIC and 1/2 MIC, respectively) increase in branching pseudohyphae (Fig. 2c). The number of chain-forming pseudohyphae was 2-fold greater when *Candida* were treated with both EOCs at their FICI. Chitin content

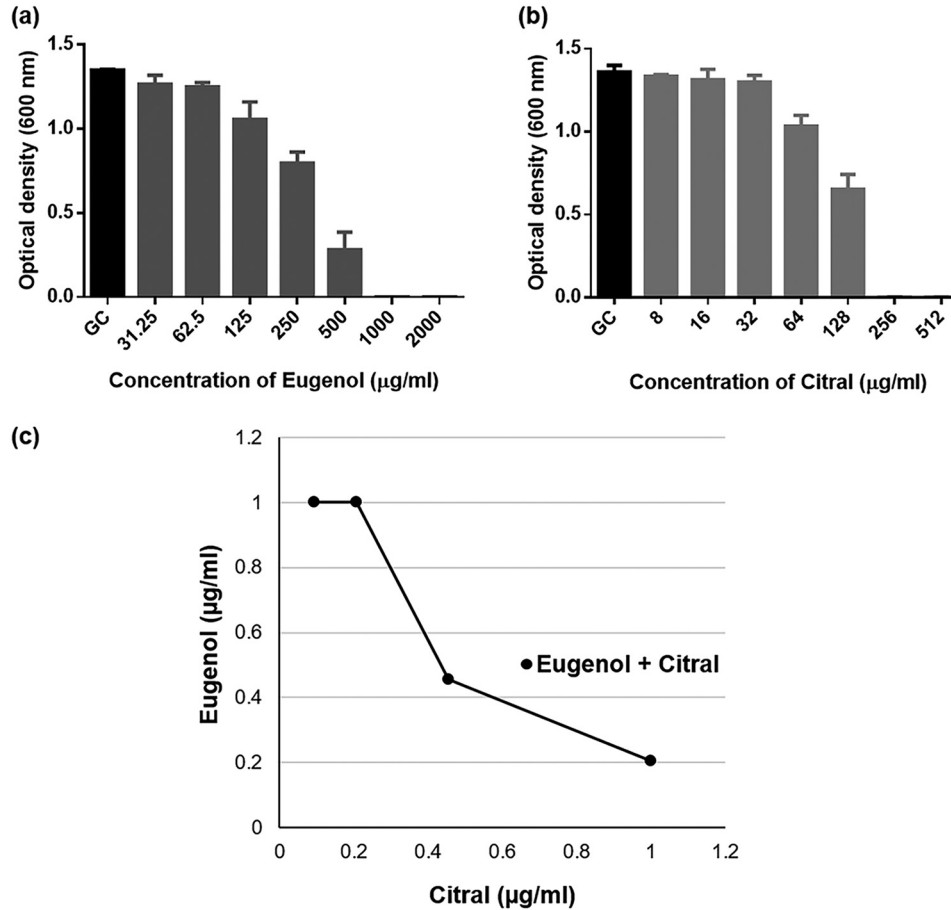

**FIG 1** Inhibitory impacts of eugenol and citral on *C. albicans* RSY150. (a and b) Bar graphs show the culture density of RSY150 exposed to eugenol (a) and citral (b) for 24 h at 30°C. GC, growth control. (c) An isobologram derived from the checker board assay shows an additive effect.

was elevated (Fig. 2b) only for eugenol and the positive control (amphotericin B [Amp B]) at MIC ($P < 0.05$ and $P < 0.01$, respectively), consistent with our previous data (41).

**Eugenol and citral induce membrane depolarization, vacuole segregation, and mitochondrial dysfunction.** Dis-C2(3) fluorescence intensity was significantly higher in *C. albicans* exposed to eugenol and citral at MIC and 1/2 MIC, or the positive control Amp B, compared to that of control (Fig. 3a and b). Eugenol and citral at their FICI also significantly ($P < 0.001$) depolarized membranes in a dose-dependent manner ($r = 0.96$ to 0.99), consistent with results for each EOC alone at 1/2 MIC (Fig. 3b).

As highlighted by pink arrows in Fig. 3c, *C. albicans* exposed to eugenol at MIC and 1/2 MIC, had a significant ($P < 0.0001$ and $P < 0.001$, respectively), concentration-dependent increase in vacuole segregation (Fig. 3b), in comparison to the large intact vacuoles observed in control cells (Fig. 3c, white arrows). However, those exposed to citral at MIC and 1/2 MIC had a mixed population of completely and partially segregated (pink and yellow arrows, respectively) vacuoles. As expected, *C. albicans* treated with either of the two EOCs or both at their FICI had significantly ($P < 0.001$) increased vacuolar segregation, most similar to that exposed to eugenol at MIC. Thus, synergy between citral and eugenol must be driving the increased vacuolar segregation (Fig. 3d).

Since EOCs are known for membrane disruption, we explored the impact of euge-nol and citral exposure on the health of the mitochondria, one of the most important membranous organelles in *C. albicans*. Untreated control cells had red, dense mito-chondria distributed throughout the cell (Fig. 3e), but the MitoTracker deep red signal was nearly abolished by treatment with eugenol at MIC or Amp B at MIC and 1/2 MIC.

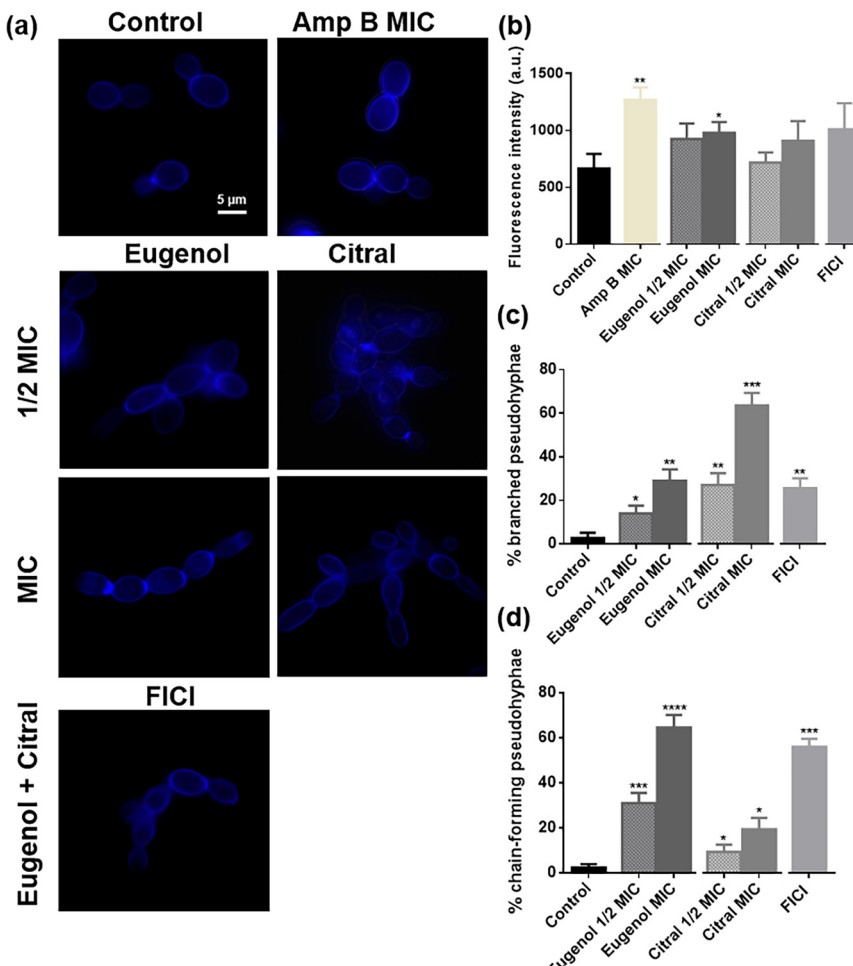

**FIG 2** Effects of eugenol and citral on *C. albicans* RSY150 morphology. (a) Epifluorescence images of calcofluor white-stained cells showed minimal impact on chitin but significant morphological changes. The scale bar for the control is 5 μm and represents the scale for all images. (b to d) Bar graphs show fluorescence intensity of all *C. albicans* RSY150 within the field of view (b), with branched (c) and chain-forming (d) pseudohyphae (respectively) after 4 h exposure to eugenol, citral, or the two combined. The data are presented as means ± the SEM of three biological replicates, with 300 cells per replicate, for which statistical significance was determined by using an unpaired Student's *t* test (****, $P < 0.0001$; ***, $P < 0.001$; *, $P < 0.05$).

Fluorescence intensity indicates a significant ($P < 0.01$) reduction in dye uptake (Fig. 3f) for EOC-treated cells compared to untreated controls. Staining of cells exposed to citral at 1/2 MIC or the EOCs at 1/2 FICI was identical to controls (Fig. 3f).

**Eugenol and citral disrupt vacuolar membrane integrity.** We further investigated whether vacuole segregation could result from loss of vacuole membrane integrity. As shown in Fig. 4a, the untreated controls had the typical ring-staining pattern of intact vacuole membranes (white arrows). In contrast, yeast vacuole marker MDY-64 was diffusely distributed in the cytoplasm of EOC-treated cells. *C. albicans* treated with eugenol at MIC and the two EOCs at their FICI had a significantly ($P < 0.001$) greater number of ruptured vacuoles (Fig. 4e), corroborating the vacuole segregation findings. As highlighted by yellow arrows in Fig. 4a, *C. albicans* exposed to citral and Amp B at MIC and 1/2 MIC, had a significant ($P < 0.0001$), concentration-dependent increase ($r = 0.98$ and 0.97, respectively) in partially ruptured vacuoles (Fig. 4d). This finding is consistent with prior studies (42, 43) and suggests a dose-dependent effect on vacuole membrane integrity.

**Eugenol- and citral-induced ROS accumulation is proportional to exposure time.** ROS accumulation can lead to *C. albicans* lethality. Figure 5a shows an increase in intracellular ROS following a 4-h exposure to the positive control, peroxide, at MIC

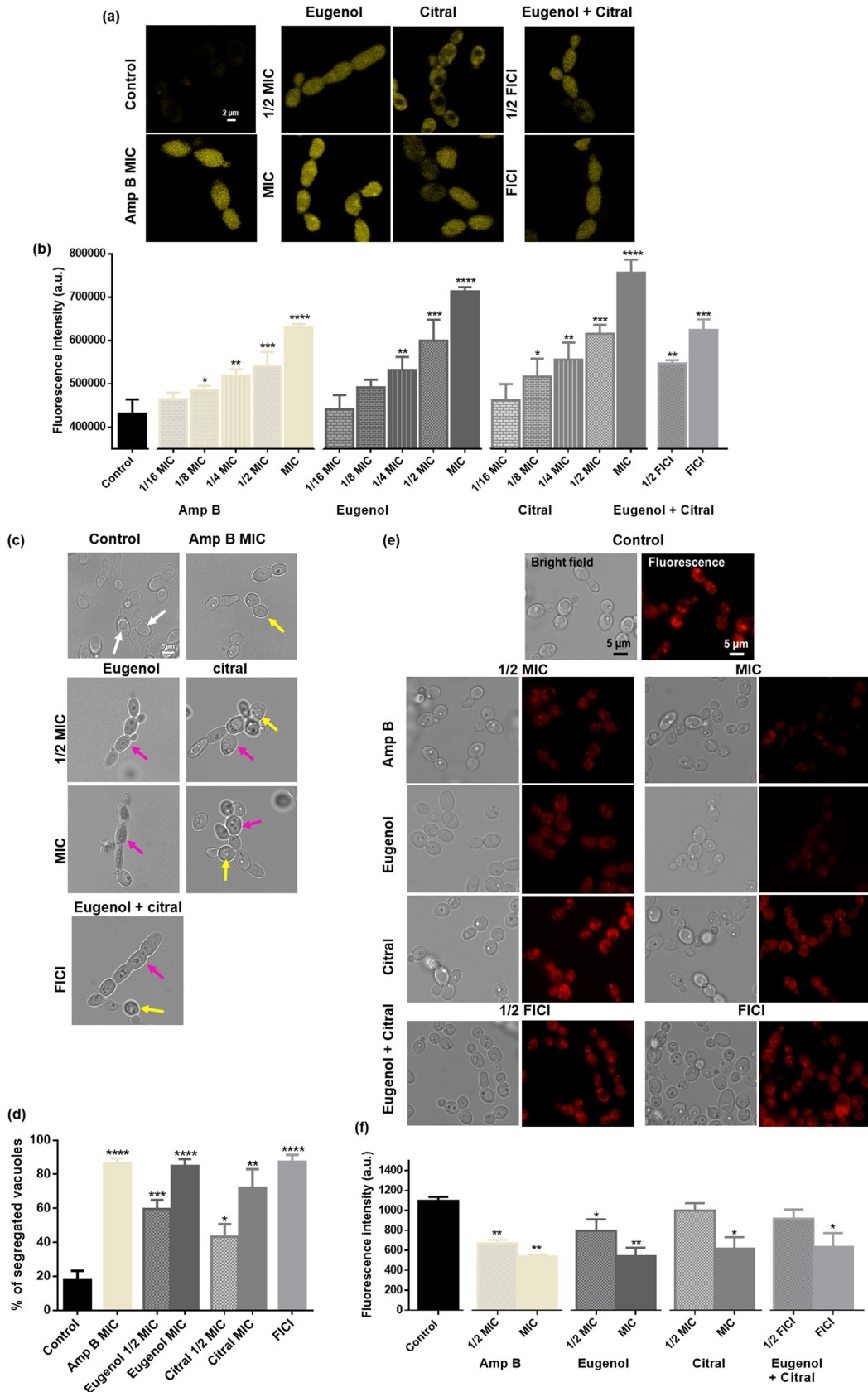

**FIG 3** Effect of eugenol and citral on the *C. albicans* RSY150 cell membrane, vacuoles, and mitochondria. (a) LSCM images show increased Dis-C2(3) dye content in treated cells compared to control. (b) Graphs showing a significant increase in *C. albicans*

and 1/2 MIC, and Amp B and eugenol at MIC. However, ROS were not generated with citral exposure nor the two EOCs at their FICI. While a 4-h citral exposure at MIC did not generate ROS, it did impact the mitochondria (Fig. 3e and f). ROS accumulation was statistically significant ($P < 0.05$) for cells exposed to eugenol at 1/2 MIC and $H_2O_2$ ($P < 0.0001$) at all concentrations (Fig. 5b). *C. albicans* exposed to citral and the two EOCs at FICI had elevated ROS (green) only after 24 h (see Fig. S2).

**Eugenol and citral exposure leads to abnormal MT structures.** Extended MTs were not observed in cells exposed to eugenol at MIC, while those exposed to EOCs at 1/2 MIC and MIC, the two at FICI or Amp B at MIC had a mixed population with short and diffuse tubulin (Fig. 6a). Delocalized tubulin indicates disruption of the mitotic spindle. The quantification of Tub2-GFP ($\beta$-tubulin) from 250 cells showed EOC-treated cells were statistically different ($P < 0.05$ and $P < 0.01$, respectively) from that of controls (Fig. 6b to d). Interestingly, cells exposed to EOCs at 1/4 MIC or the two EOCs at their 1/4 FICI had a significantly ($P < 0.05$) greater number (Fig. 6b) of short tubulin clusters (small fluorescent spots) or diffuse tubulin (Fig. 6a), respectively. Exposure to the known $\beta$-tubulin inhibitor, nocodazole, at full, 1/2 and 1/4 MIC resulted in a significant ($P < 0.01$ and $P < 0.05$, respectively) increase in short tubulin (Fig. 6a to d).

**Eugenol and citral cause cell cycle arrest in *C. albicans*.** To determine whether cell membrane depolarization, vacuole defects and abnormal MTs were associated with cell cycle arrest, the different cell cycle stages were quantified from LSCM images. For the RSY150 control cells, 43% were in $G_1$ or S phase, 12% had undergone mitosis, and the remaining 45% were in the postmitotic phase (Fig. 6e, control). Exposure to citral or eugenol at MIC or the two at their FICI led to significant ($P < 0.01$, $P < 0.001$, $P < 0.01$) changes in cell cycle distribution, with 70, 69, and 64%, respectively, not having begun mitosis which indicates cell cycle arrest at the $G_1$/S phase (Fig. 6e).

**Eugenol and citral cause cell death.** An intracellular content leakage assay ($A_{260}$) was used to assess the degree of membrane rupture for *C. albicans* RSY150, showing a concentration (1/16 MIC to MIC) dependent increase ($r = 0.96$ to $0.97$) for treated cells (Fig. 7a). RSY150 exposed to citral (MIC, 1/2 MIC) had significantly ($P < 0.0001$) higher leakage than those exposed to eugenol or the EOCs at FICI, whereas the positive control, Amp B, caused significant ($P < 0.001$) leakage only at MIC (Fig. 7a).

AFM images of *C. albicans* RSY150 exposed for 4 h to Amp B, eugenol and citral at 1/4 MIC, 1/2 MIC, and MIC, the two EOCs at 1/2 and full FICI had surface ultrastructural changes (Fig. 7b) that were accompanied by swelling and possible leakage. Only RSY150 exposed to Amp B at MIC showed this effect, with increased surface roughness at 1/2 and 1/4 MIC (Fig. 7b). The height of *C. albicans* RSY150 (Fig. 7c) significantly increased (Table 1) after exposure to Amp B, eugenol, and citral at MIC and the two at FICI, respectively (Fig. 7c), but there was no change when exposed to EOCs at 1/2 and 1/4 MIC compared to untreated controls. Control cells had smooth and homogeneous surfaces with a roughness of $5.7 \pm 1.2$ nm, which significantly increased (Table 1) when treated with both Amp B and EOCs at their MIC and 1/2 MIC (Fig. 7d).

To confirm that eugenol and citral alter *C. albicans* membrane integrity, we used the cell membrane impermeable fluorescent dye propidium iodide (PI) which is unable to enter intact cells. *C. albicans* RBY1132 controls (see Fig. 7e, Fig S3) had no PI fluorescence,

**FIG 3** Legend (Continued)

RSY150 membrane depolarization with exposure to EOCs at MIC, 1/2 MIC, or both EOCs at FICI compared to controls. Pearson correlation indicated a positive association (eugenol, $r = 0.96$, $P < 0.001$; citral, $r = 0.96$, $P < 0.01$; FICI, $r = 0.99$, $P < 0.01$; Amp B, $r = 0.96$, $P < 0.001$) between oil concentration and membrane depolarization. (c) Eugenol, citral, and the both at FICI caused vacuolar segregation (pink arrows) compared to control (white arrows). Amp B at MIC showed partial (yellow arrow) segregation compared to EOC-treated cells. (d) Bar graphs showing the percentage increase in segregated vacuoles for treated cells compared to controls. (e) Bright-field (left) and fluorescence (right; $\lambda_{ex} = 644$ nm; $\lambda_{em} = 665$ nm) images of RBY1132 exposed to eugenol and citral at MIC and the two at their full FICI showed poor MitoTracker deep red uptake compared to controls. (f) Bar graphs of MitoTracker deep red fluorescence intensity show significantly fewer active mitochondria for *C. albicans* treated with EOCs compared to controls. Scale bars are 2 $\mu$m for panel a and 5 $\mu$m for panels c and e. The data are presented as means $\pm$ the SEM of three biological replicates, for which statistical significance was analyzed by a one-way ANOVA, followed by a Dunnett's multiple comparison of each condition versus the control for panel b and an unpaired Student's $t$ test for panels d and f (****, $P < 0.0001$; ***, $P < 0.001$; **, $P < 0.01$; *, $P < 0.05$).

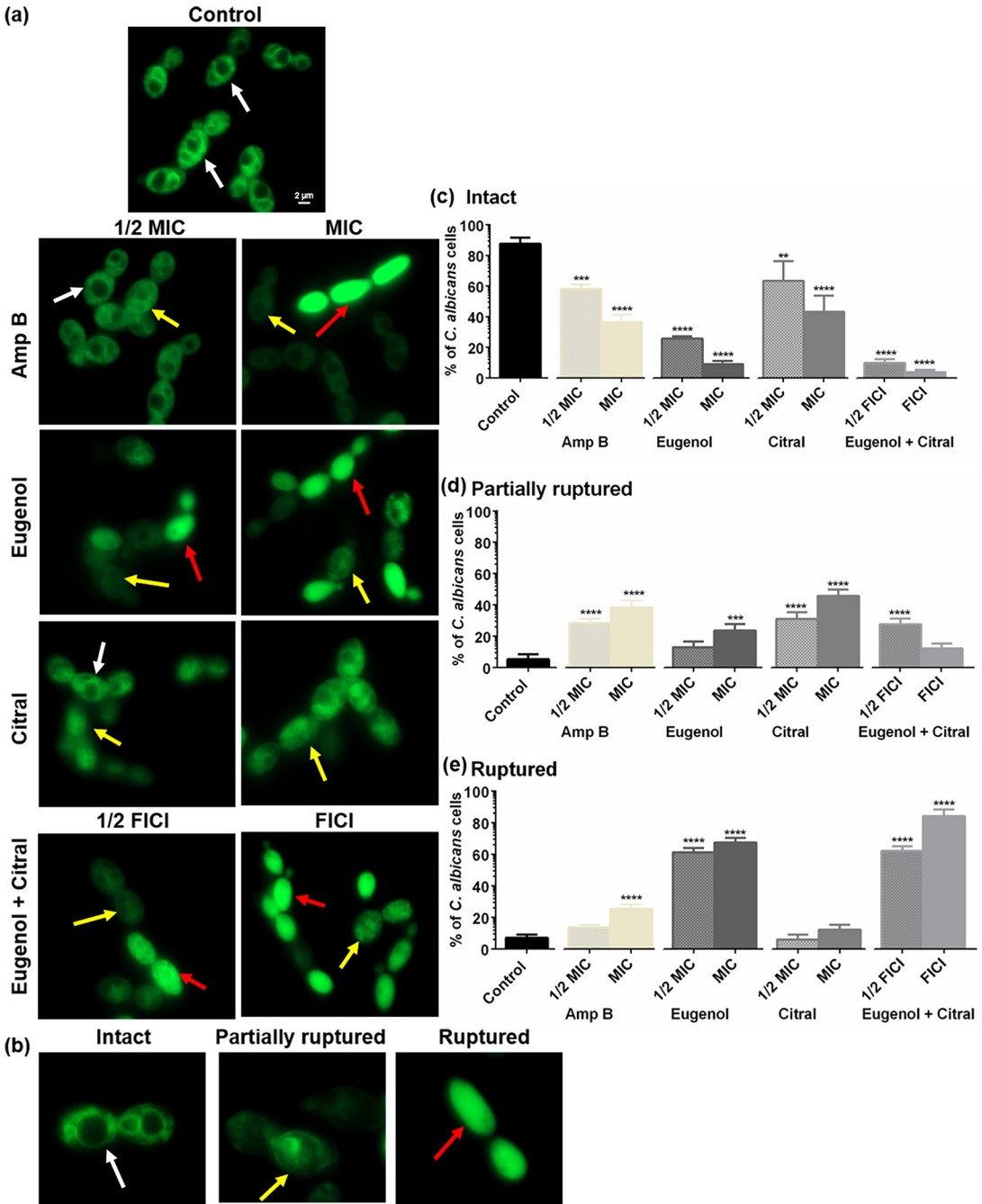

**FIG 4** Impact of eugenol and citral on *C. albicans* RSY150 vacuole membrane integrity. (a) Epifluorescence images of MDY-64 ($\lambda_{ex}$ = 451 nm; $\lambda_{em}$ = 497 nm)-stained cells showed an impact on vacuole membranes after 4 h of EOC treatment. The scale bar for the control is 2 $\mu$m and represents the scale for all images. (b) The white, yellow, and red arrows represent intact, partially ruptured, and ruptured vacuole membranes, respectively. (c to e) Bar graphs determined from a quantitative analysis of the percentages (%) of *C. albicans* cells having intact, partially ruptured, and ruptured vacuolar membranes. The data are reported as means $\pm$ the SEM of three biological replicates, for which statistical significance was evaluated by one-way ANOVA, followed by Dunnett's multiple comparison of each condition versus the control (****, $P < 0.0001$; ***, $P < 0.001$).

which was significantly ($P < 0.001$) increased with exposure to the positive control, $H_2O_2$, eugenol at MIC, and slightly less so ($P < 0.01$) for citral at MIC (see Fig. 7e, Fig. S3). On the other hand, exposure to citral at 1/2 MIC or the two EOCs at FICI showed significant ($P < 0.05$) PI uptake, but less than that of the positive control and eugenol at 1/2 MIC.

**Eugenol and citral hinder hyphal and biofilm formation in *C. albicans*.** Approximately 89% of the untreated *C. albicans* Tub2-GFP expressing strain formed germ tubes with

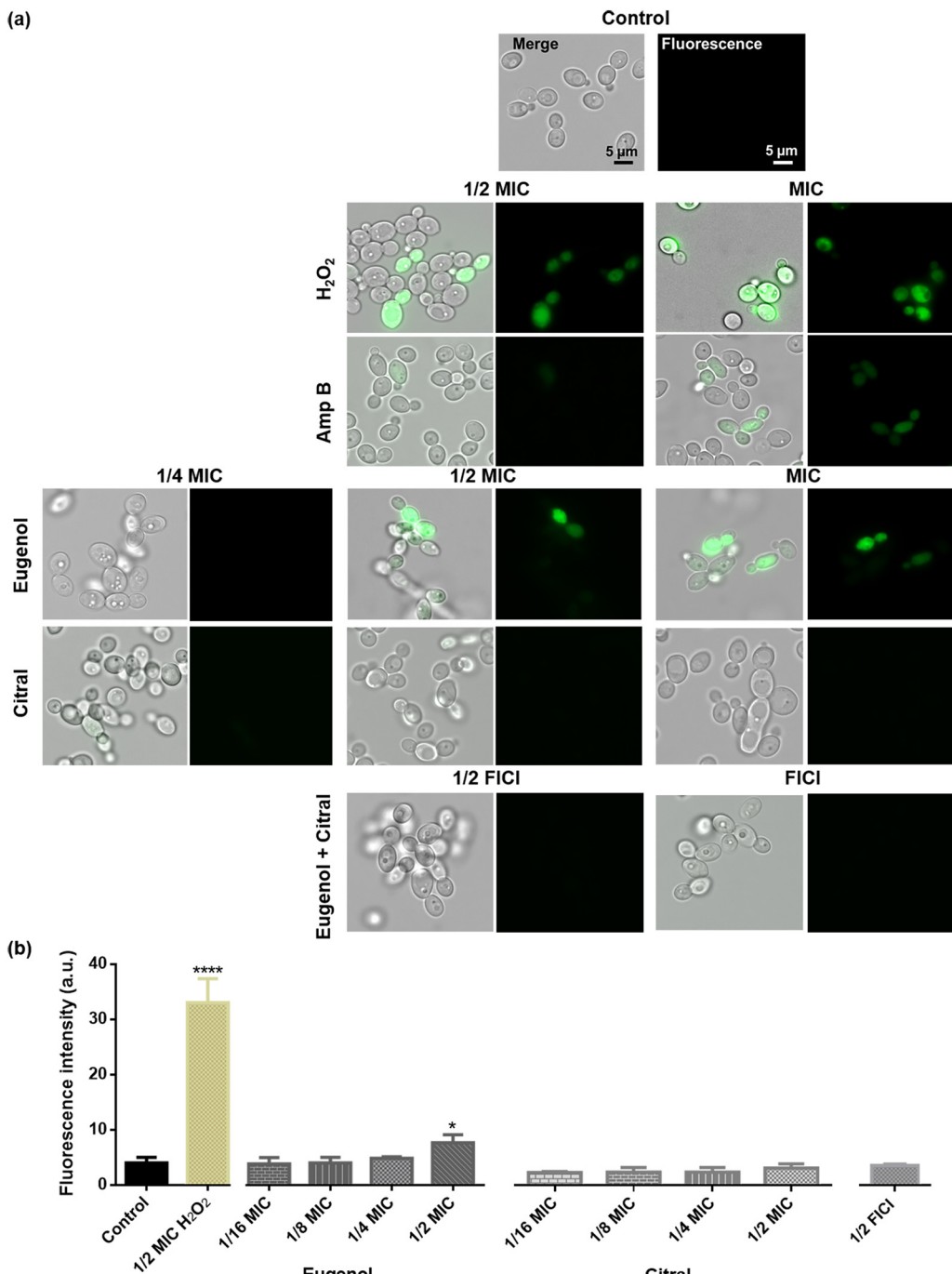

**FIG 5** Impact of eugenol and citral on *C. albicans* RBY1132 intracellular ROS accumulation. (a) Merged (bright-field/ fluorescence; left) and fluorescence (right) images show ROS content in *C. albicans* RBY1132 treated with eugenol at MIC and 1/2 MIC, absent in the control and those treated with citral or the two at their FICI. Scale bars are 5 $\mu$m for controls and are representative of all images. (b) Bar graphs show a significant ROS signal for *C. albicans* exposed to eugenol and $H_2O_2$ (25 mM) at 1/2 MIC. Fluorescence intensity of the ROS indicator was measured in a plate reader ($\lambda_{ex}$ = 485 nm; $\lambda_{em}$ = 528 nm, gain 35). The data are presented as means ± the SEM of three biological replicates for which statistical significance was evaluated by a one-way ANOVA, followed by a Dunnett's multiple comparison of each condition versus the control (****, $P < 0.0001$; *, $P < 0.05$).

regular MTs, but in the presence of EOCs (Fig. 8b) had diffuse tubulin (green fluorescence) and significantly fewer ($P < 0.0001$) germ tubes. RSY150 treated with citral at 1/2 MIC and Amp B at MIC and 1/2 MIC had a statistically significant ($P < 0.001$) reduction (23%) in germ tube formation, accompanied by hyphal developmental deficiency

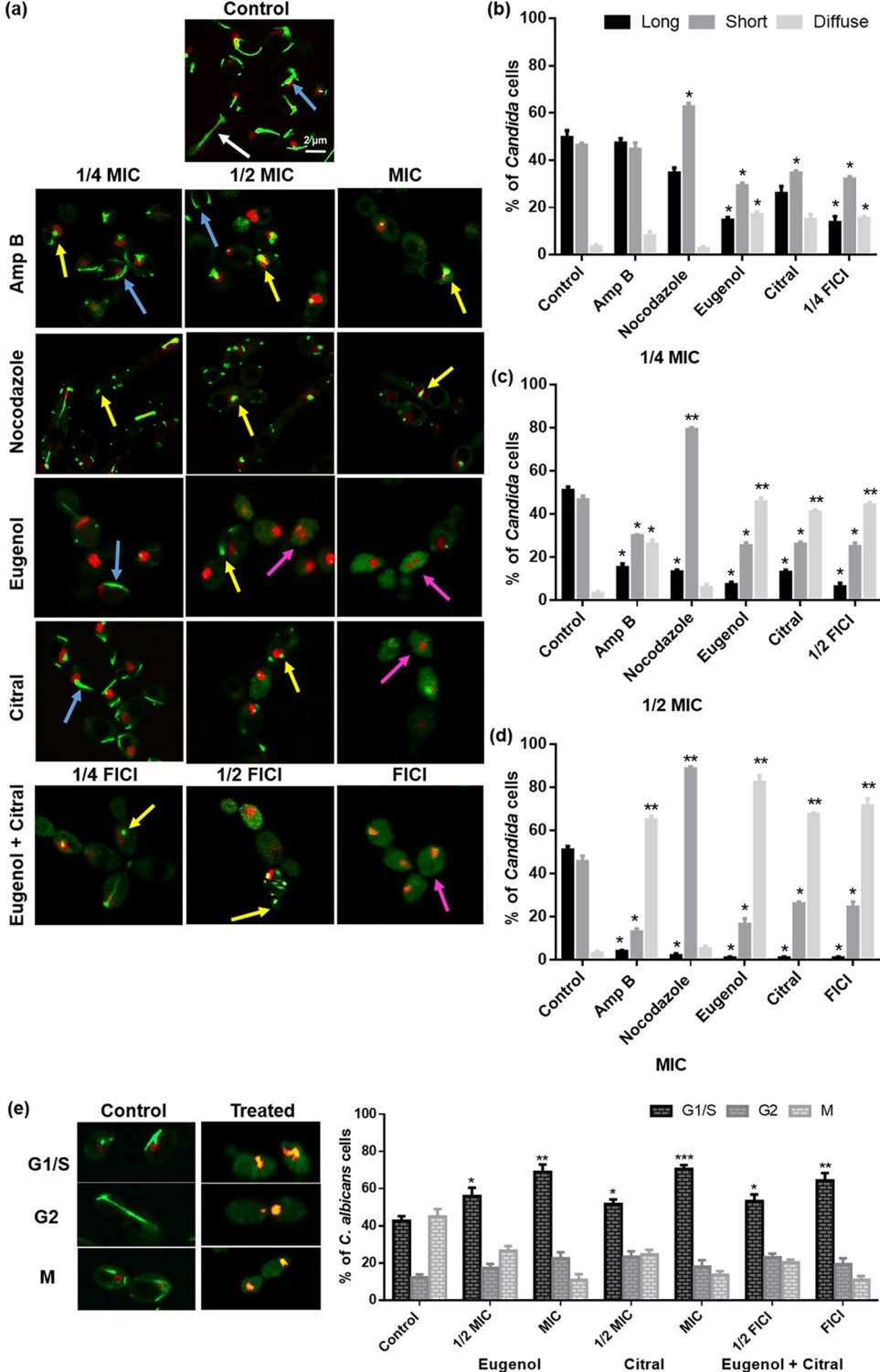

**FIG 6** Effects of eugenol and citral on *C. albicans* RSY150 MT formation and cell cycle. (a) Representative images collected by LSCM (Tub2-GFP, $\lambda_{ex}$ = 488 nm, $\lambda_{em}$ = 512 nm; Htb-RFP, $\lambda_{ex}$ = 543 nm; $\lambda_{em}$ = 605 nm) show the majority of cells treated with eugenol or citral at MIC and the two at their FICI having diffuse tubulin (green fluorescence, purple arrows) compared to untreated controls showing normal spindles during mitosis (green fluorescence, white and blue arrows indicate long and short MTs, respectively). Cells treated with the positive control, nocodazole, at 1/2 MIC and MIC had concentrated spots of green fluorescence (yellow arrow), similar to EOC-treated cells at 1/4 and 1/2 MIC, but distinct from the short spindles (blue arrow) observed in controls and Amp B-treated (1/4 and 1/2 MIC) cells. The scale bar for the control is 2 $\mu$m and represents the scale for all images. (b to d) Bar graphs show the change in $\beta$-tubulin morphology with EOC exposure. (e) Cell cycle phase (S/G$_1$, G$_2$, or M) distributions were counted from images, in which untreated controls had well-distributed MTs,

(Fig. 8a and b). All treated cells produced pseudohyphae (chain and cluster), consistent with the morphology analysis.

*C. albicans* RSY150 in preformed biofilm were much less viable in the presence of eugenol, citral, and the two at their FICI (Fig. 8c; see also Fig. S4), an effect that was concentration-dependent ($r$ = 0.91 to 0.96). Exposure of *C. albicans* to EOCs at MIC significantly ($P < 0.0001$) hindered biofilm formation, which was less prominent but significant ($P < 0.001$ and $P < 0.01$, respectively) with exposure to EOCs from 1/4 to 1/8 MIC (Fig. 8c). RSY150 exposed to EOCs at 1/16 MIC were statistically identical to untreated cells.

*C. albicans* RSY150 constantly exposed to eugenol and citral at different fractional MICs, and the two at their FICI, produced fewer mycelia. When grown on spider media, colonies of treated cells appeared smooth and round, whereas, control colonies were wrinkled and uneven (Fig. 8d; see also Fig. S5, control). Filaments (hyphal growth) at the edges of control colonies were absent from those exposed to eugenol at 1/2 and 1/4 MIC, citral at 1/2 MIC and the two at 1/2 FICI, with the latter having very smooth, wrinkle-free colony surfaces (Fig. 8d; see also Fig. S5). *C. albicans* preincubated for 4 h with eugenol at 1/16 to 1/2 MIC, citral at 1/4 to 1/2 MIC, or the two at 1/2 FICI, and further incubated for a total of 6 days after the removal of EOCs had reduced mycelial growth compared to untreated controls (see Fig. S6a and b). The latter indicates that EOC exposure permanently inhibits certain cellular processes that prevent *Candida* returning to its fully invasive form over a number of days (see Fig. S5c). Interestingly, pretreatment with Amp B at 1/2 MIC failed to inhibit mycelium formation.

## DISCUSSION

The cell wall and cell membrane are essential not only for *Candida* cell viability and morphology, but also virulence through several signaling pathways (44). Therefore, relatively slight changes to the structural integrity of the cell wall (41) or membrane can result in cell death. Eugenol and citral are both lethal for *C. albicans* at relatively high concentrations compared to commercial antifungals (45) and, together, their anticandidal activity is additive (Fig. 1). Cell swelling and leakage (Fig. 7) was indicative of severe membrane disruption (see Fig. S3) and EOC-induced lethality at high concentrations (Fig. 9). Eugenol and citral both impact the cell membrane (17, 25) by embedding in the lipid bilayer and disrupting its fluidity and permeability (46, 47). Membrane depolarization (Fig. 2) triggers an inflammatory response that perpetuates cytoplasmic leakage (18, 27, 48), as observed in this study (Fig. 7). Ions play a vital role in *C. albicans* as cofactors for multiple enzymes, by maintaining membrane potential, regulating cell volume, along with roles in proliferation and apoptosis (49), and therefore the swelling and leakage may result from improper ion balance following EOC exposure. Eugenol's phenolic constituent (–OH) has been proposed to enable hydrogen-bonding and proton exchange, cause electron delocalization and membrane perturbations that impact enzyme action (14–16), and aliphatic aldehydes like citral are capable of forming a charge transfer complex with tryptophan (50, 51). Thus, depolarization of the *C. albicans* cell membrane by eugenol and citral alone and in combination (Fig. 3) is consistent with previous studies (17, 26, 27, 31).

EO(C)-induced disruption of membrane integrity can lead to cell cycle arrest (30), for example with linalool and citral (30). The $G_1$/S arrest observed in this study (Fig. 6) is consistent with eugenol and citral arresting *C. albicans* at the S phase (29, 30). Indeed, *S. cerevisiae* cell membrane damage leads to transient cell cycle arrest in $G_1$ (52). Genetic defects having an acute block in vacuole biogenesis generate nonfunctional vacuoles with cell cycle arrest at the early $G_1$ phase (53), setting a precedent for

**FIG 6** Legend (Continued)
but those treated with EOCs were mostly in the $G_1$/S phase with defective or absent MTs. Bar graphs are presented as means ± the SEM of three biological replicates, with 250 cells per replicate, for which statistical significance (***, $P < 0.001$; **, $P < 0.01$; *, $P < 0.05$) was evaluated by a one-way ANOVA, followed by Dunnett's multiple comparison of each condition versus the control (b to d) or by an unpaired Student's *t* test (e).

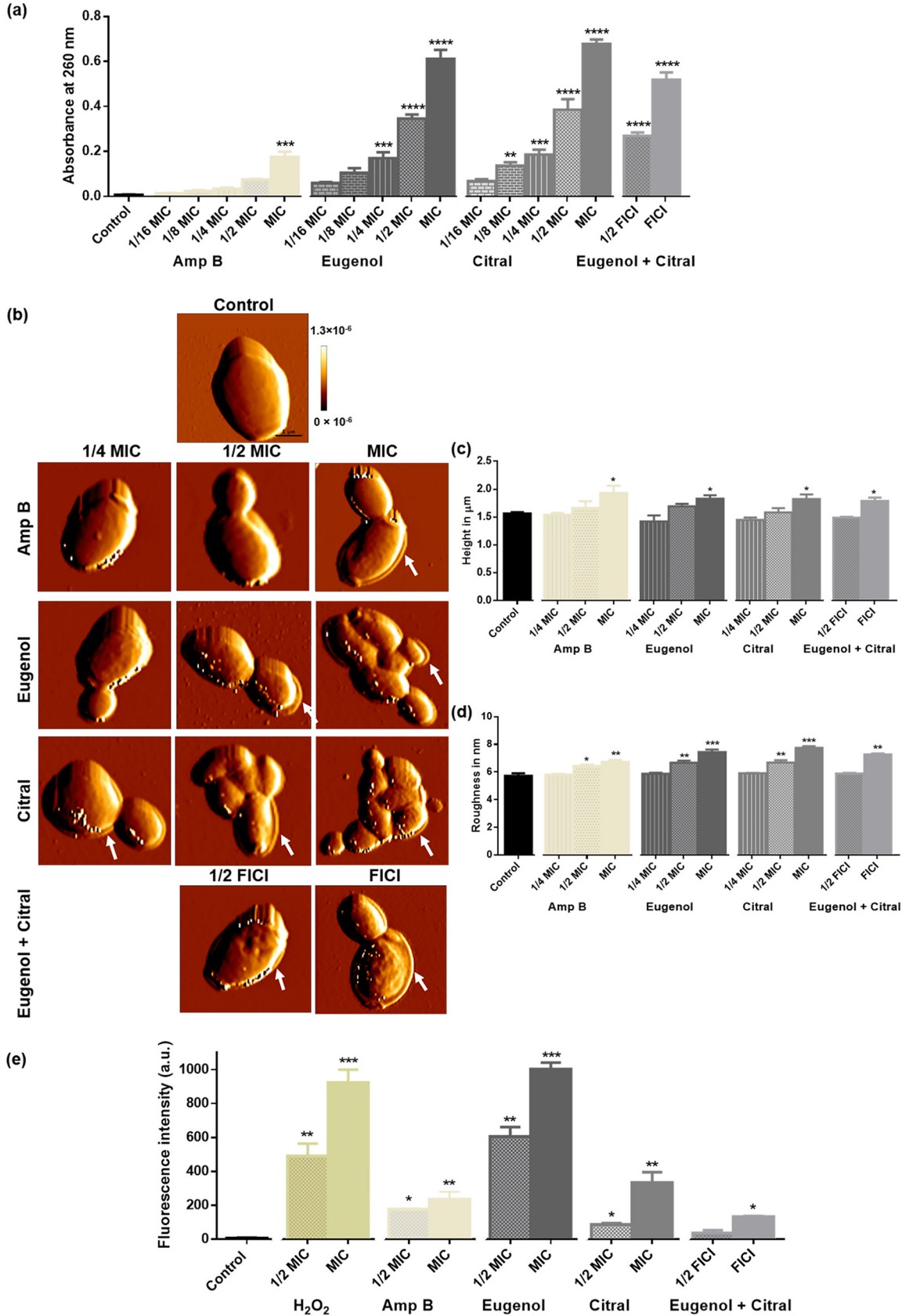

**FIG 7** *C. albicans* RSY150 exposed to eugenol and citral showed swelling, leakage, membrane damage and cell death. (a) Exposure to eugenol, citral, and Amp B caused leaky membranes in *C. albicans*. The Pearson correlation indicates a positive association (eugenol, *r* = 0.96,

**TABLE 1** *C. albicans* RSY150 height and roughness after exposure to EOCs

| Parameter and treatment | Mean Height ($\mu$m) or roughness (nm) ± SEM[a] | | |
|---|---|---|---|
| | **1/4 MIC** | **1/2 MIC** | **MIC** |
| **Height ($\mu$m)** | | | |
| Control | 1.57 ± 0.02 | 1.57 ± 0.02 | 1.57 ± 0.02 |
| Amp B | 1.54 ± 0.03 | 1.67 ± 0.12 | 1.93 ± 0.13* |
| Eugenol | 1.43 ± 0.11 | 1.69 ± 0.04 | 1.83 ± 0.06* |
| Citral | 1.45 ± 0.04 | 1.59 ± 0.08 | 1.83 ± 0.08* |
| | | **1/2 FICI** | **FICI** |
| Eugenol + citral | | 1.49 ± 0.02 | 1.79 ± 0.06* |
| | **1/4 MIC** | **1/2 FICI** | **FICI** |
| **Roughness (nm)** | | | |
| Control | 5.73 ± 1.18 | 5.73 ± 1.18 | 5.73 ± 1.18 |
| Amp B | 5.82 ± 0.05 | 6.45 ± 0.10* | 6.74 ± 0.14** |
| Eugenol | 5.88 ± 0.07 | 6.68 ± 0.13** | 7.45 ± 0.18*** |
| Citral | 5.92 ± 0.01 | 6.69 ± 0.16** | 7.76 ± 0.11*** |
| | | **1/2 FICI** | **FICI** |
| Eugenol + citral | | 5.89 ± 0.05 | 7.28 ± 0.07** |

[a]***, $P < 0.001$; **, $P < 0.01$; *, $P < 0.05$.

the observed vacuole segregation and loss of vacuolar membrane integrity (Fig. 3 and Fig. 4). Similar findings have been reported for *C. albicans* exposed to known membrane disruptors such as Amp B (41), cationic peptides (54), clove, thyme (55) rosemary and its major components (41).

The mechanism of EOCs is not completely established, but early studies show the trend of ROS-induced mitochondrial outer membrane permeability which allows the release of *cytochrome c* and other proapoptotic factors, leading to subsequent metacaspase activation and ultimately apoptosis (20, 56). This sequence is consistent with the impact of eugenol on *C. albicans* (Fig. 5). While oxidative stress plays an important mechanistic role for eugenol-induced cell death, the impact of citral is more complicated. Both eugenol and citral at MIC altered mitochondrial membrane potential in *C. albicans* to a similar degree (Fig. 3), but citral failed to generate ROS, consistent with previous reports for *A. flavus*, *Trichophyton mentagrophytes*, and *Tagetes patula* (57, 58). On the other hand, limited exposure of *P. digitatum* to citral led to abnormal mitochondria, reduced respiration (58) and ATP production, accompanied by intracellular ROS (59). The oxidation of tubulin sulfhydryl groups by ROS (60, 61) would prevent *C. albicans* MT assembly, and certainly $H_2O_2$-generated ROS has been shown to deteriorate MT dynamics (60). However, eugenol and citral also caused MT defects at low fractional MICs (Fig. 6), at which ROS levels are not elevated (Fig. 5). Citral disrupts MTs in plant and animal cells, with no damage to the cell membrane nor the actin cytoskeleton, but plant cells exposed to limonene and (+)-citronellal have membrane damage associated with MT disruption (62, 63), consistent with this study and similar to the impact of other EO(C)s (41, 64).

It would be tempting to postulate that exposure to EOCs at higher concentrations or longer times results in mitochondrial damage and consequent ROS production, causing further cell membrane damage, cytoplasmic leakage, and cell death (Fig. 9). Indeed, Amp B induces an oxidative burst in *Cryptococcus neoformans* and disruption of the cytoplasmic membrane (65), proposed to be mediated by ergosterol binding

**FIG 7** Legend (Continued)

$P < 0.001$; citral, $r = 0.97$, $P < 0.001$; FICI, $r = 0.96$, $P < 0.001$) between the oil concentration (1/16 MIC, 1/8 MIC, 1/4 MIC, 1/2 MIC, and MIC) and partial cell collapse. (b) Representative low-resolution (10 $\mu$m, 128 × 128 pixel) AFM images of untreated *C. albicans* and those exposed to Amp B, eugenol, and citral (1/4 MIC) and the two at FICI (1/2, full) showed swelling and possible leakage (white arrows) after EOC treatment. The scale bar is 1 $\mu$m for the control and represents the scale for all images. (c and d) Bar graphs show changes to cell height (c) and surface roughness (d) from AFM images of 20 cells for each of two biological replicates. (e) Bar graphs of the PI intensity show treated cells with increased cell membrane damage compared to control. The data are presented as means ± the SEM of three biological replicates, for which statistical significance (****, $P < 0.0001$; ***, $P < 0.001$; **, $P < 0.01$; *, $P < 0.05$) was evaluated by one-way ANOVA, followed by a Dunnett's multiple comparison of each condition versus control (a, c, and d) or an unpaired Student's *t* test (e).

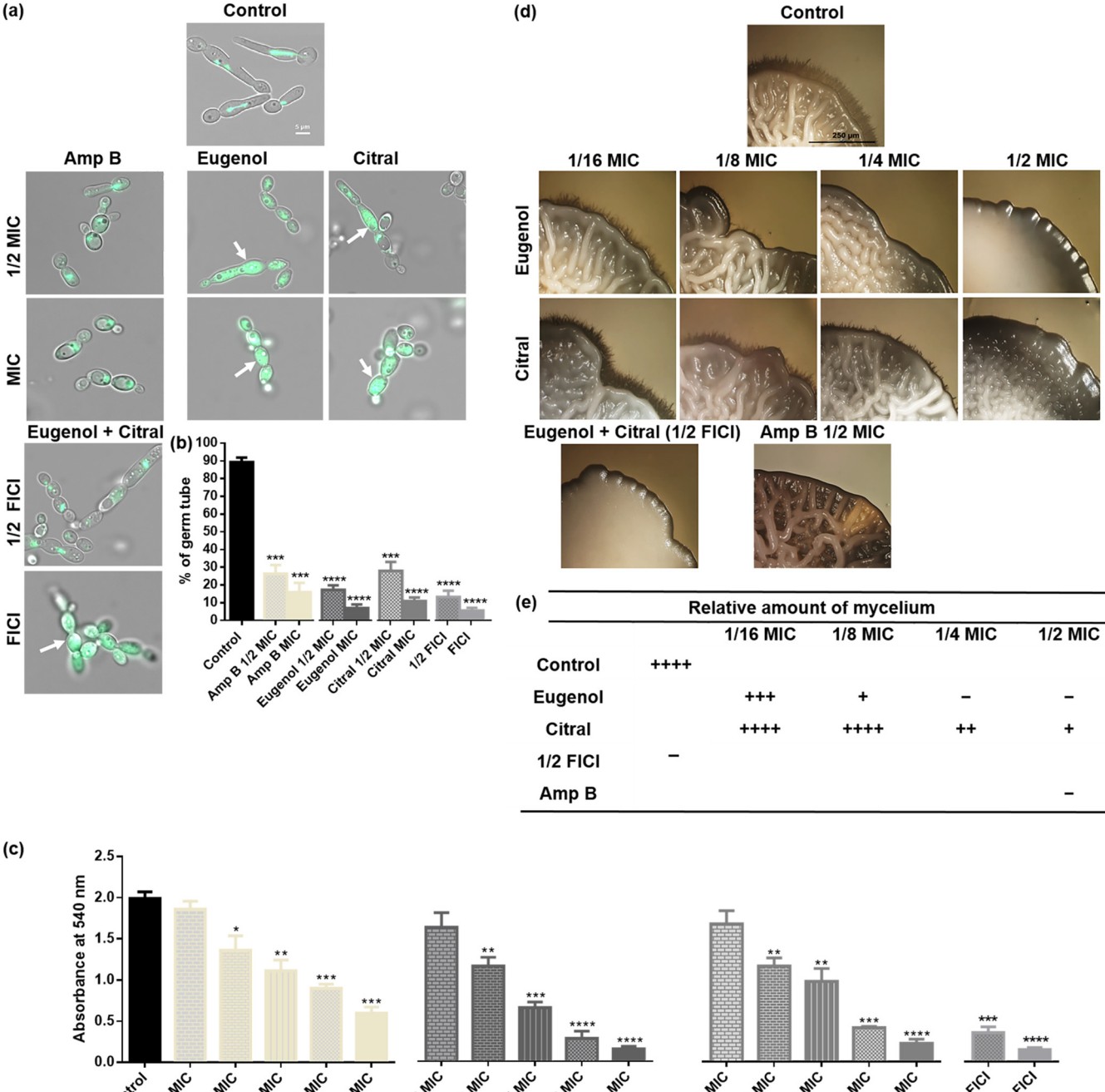

**FIG 8** Impact of eugenol and citral on *C. albicans* RSY150 morphological switching and biofilm formation. (a) Representative merged bright-field and fluorescence (Tub2-GFP, $\lambda_{ex}$ = 488 nm, $\lambda_{em}$ = 512 nm) microscopy images of *C. albicans* treated with 10% FBS in YPD medium containing eugenol, citral (1/2 MIC and MIC), or the two at 1/2 and full FICI (4-h exposure) show pseudohyphae with diffuse $\beta$-tubulin (white arrows). The scale bar is 5 $\mu$m for the control and is applicable to all images. (b) Bar graphs show the significant impact of EOCs on germ tube formation, with 100 cells per replicate (****, $P < 0.0001$; ***, $P < 0.001$). (c) MTT assay results show significant (****, $P < 0.0001$; ***, $P < 0.001$; **, $P < 0.01$) and dose-dependent (eugenol, $r = 0.94$, $P < 0.001$; citral, $r = 0.96$, $P < 0.001$; citral/eugenol [FICI], $r = 0.91$, $P < 0.01$; Amp B, $r = 0.92$, $P < 0.01$) reductions in preformed biofilm after exposure to EOCs (1/8 MIC to MIC), which is particularly prominent for eugenol at MIC and 1/2 MIC, citral at MIC, and both at FICI. (d) Representative bright-field images of spider medium agar plates show colony morphology (scale bar, 250 $\mu$m, applicable to all images) constantly exposed to eugenol and citral (6 days), for which the symbols in panel e (+, ++, +++, and ++++) indicate the relative amounts of mycelial growth. The data are presented as means ± the SEM of three (b) and four (c) biological replicates, for which statistical significance was analyzed by an unpaired Student's *t* test or a one-way ANOVA, followed by Dunnett's multiple comparison of each condition versus the control, respectively.

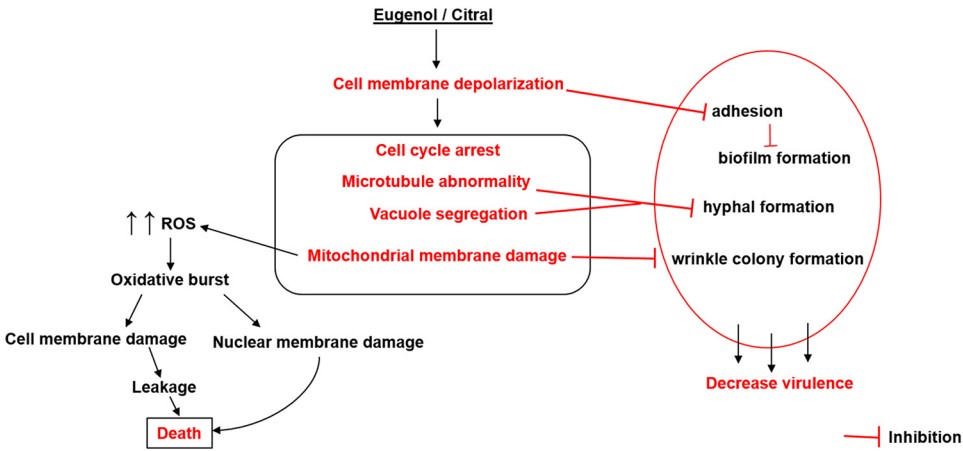

**FIG 9** Model for the two-tier anticandidal action of eugenol and citral. We postulate that both EOC concentration and exposure lead to different cell penetration. At higher EOC concentrations, after saturation and depolarization of the cell membrane, EOCs can access organelle membranes and cytosolic proteins. Penetration into the mitochondrial membrane contributes to ROS and oxidative bursts, which further cause damage to essential cellular components (e.g., nucleus, microtubules), ultimately resulting in cell death. On the other hand, defects in the cell membrane, vacuoles, mitochondria, and MTs at low EOC concentrations hamper important virulence factors and therefore affect pathogenicity.

(66) and formation of transmembrane pores (67) that allow leakage of cytoplasmic contents (68). However, Ferreira et al. (69) suggest that Amp B-induced cell death is not a simple consequence of changes to cell membrane permeability, consistent with this study. Interestingly, the eugenol phenolic (–OH) group can lose its proton easily (16), resulting in increased membrane fluidity and permeability, inhibition of respiration and alteration of ion transport processes (70, 71). This would explain why a short eugenol exposure can induce ROS in *C. albicans*, while citral requires longer exposure times (see Fig. S2).

The yeast-to-hyphal transition and biofilm formation are vital factors for *C. albicans* pathogenicity, and interestingly both were impaired with exposure to eugenol and citral (Fig. 8; see also Fig. S4 to S6) in a ROS-independent manner. *C. albicans* hyphal formation not only plays important roles in host tissue invasion, but is also essential for biofilm formation (72). The dose-dependent reduction in biofilm (Fig. 8; see also Fig. S4) is consistent with prior studies showing how eugenol inhibits sessile cells in *C. albicans* biofilms and invasive growth (21, 23, 29, 73). *C. albicans* vacuoles (74) and the MT cytoskeleton are tightly coordinated with morphology (75) and are essential regulators for hyphal growth (76). Thus, it is conceivable that their disruption (Fig. 4 and Fig. 6) represents a potent route to impair hyphal formation and morphological switching at sublethal EOC exposure, where ROS levels are statistically identical to unexposed *Candida* (Fig. 6). This idea is supported by previous studies suggesting that *C. albicans* mitochondrial respiration is linked to filamentation and biofilm development (77–79), with chemicals impacting mitochondrial activity also inhibiting morphological transitions (77–80). Indeed, with exposure to EOCs at sublethal levels *C. albicans* takes on a wrinkleless colony phenotype (Fig. 8; see also Fig. S5 and S6) that has been attributed to mitochondrial dysfunction (41).

Although several studies have reported compounds with potent activity against *C. albicans* mycelial, hyphal and biofilm growth (27, 28, 40), this is the first report showing a ROS-independent impact on vacuoles, mitochondria and MTs (Fig. 9) at sublethal EOC concentrations. Given the high concentration of EOCs required for *C. albicans* lethality, their use at low fractional MIC to inhibit *C. albicans* virulence presents an interesting area for future study.

## MATERIALS AND METHODS

**Strains and growth conditions.** *C. albicans* RBY1132 and RSY150 (81) were kindly provided by Richard J. Bennett (Department of Molecular Microbiology and Immunology, Brown University, Providence, RI). The isolates were stored in 50% glycerol stock at −80°C until required.

**Essential oil components and media.** Bacto agar, yeast extract, and peptone were obtained from Difco (BD Biosciences, Franklin Lakes, NJ), and Amp B (amphotericin), nocodazole, fetal bovine serum, CaCl$_2$, glucose, K$_2$HPO$_4$, NaCl, mannitol, nutrient broth, phosphate-buffered saline powder (PBS; 0.01 M phosphate [pH 7.4], 0.138 M NaCl, and 0.0027 M KCl in 1 L of ultrapure water) were purchased from Sigma-Aldrich Chemical Co. (St. Louis, MO). Eugenol (99%) and citral (95%) were purchased from Acros Organics (Morris Plains, NJ) and Sigma-Aldrich (St. Louis, MO), respectively. Stock solutions (8,000 and 2,048 $\mu$g/mL for eugenol and citral, respectively) were prepared using 0.2% Tween 80, which was effective for solubility with no impact on fungal growth. Prior to each experiment, *C. albicans* strains were freshly revived on yeast peptone dextrose (YPD) media containing 1% yeast extract, 2% peptone, and 2% glucose.

Cells at log phase were diluted with YPD broth to 10$^5$ cells/mL (optical density at 600 nm [OD$_{600}$] = 0.001, equivalent to 1.2 $\times$ 10$^5$ cells/mL or a 0.5 McFarland standard) for MIC and checkerboard assays (82) and $\sim$10$^7$ CFU/mL from the mid-log phase were used for all other experiments as described by Shahina et al. (64). Unless otherwise stated, *C. albicans* overnight cultures were treated with EOC(s) for 4 h with shaking (200 rpm) at 30°C.

**MIC and FICI assays.** The MICs of eugenol and citral preventing growth (OD$_{600}$) were determined using the broth micro dilution method according to Clinical and Laboratory Standards Institute (CLSI) guidelines (82).

The broth microdilution checkerboard assay was used to determine the fractional inhibitory concentration index (FICI), calculated as the sum of each EOC MIC in combination divided by the MIC of the EOC alone. FICIs of ≤0.5 were interpreted as synergistic, 0.5 < FICI ≤ 0.75 values were interpreted as partially synergistic, 0.76 < FICI ≤ 1.0 values were interpreted as additive, while values corresponding to 1.0 < FICI ≤ 4.0 and FICI > 4.0 were considered indifferent or antagonistic (83). Isobolograms were constructed by plotting the dose of EOC 1 versus that of EOC 2, for which a straight line connecting the intercept points represents zero interaction, a concave curve below that line is considered synergistic (FICI ≤ 0.5) or additive (0.5 < FICI < 1), while a convex curve above the line antagonistic (FICI > 4) (84). For comparison purposes, essential oil concentrations are expressed throughout as fractional (1/8, 1/4, and 1/2) MIC and fractional (1/8, 1/4, and 1/2) FICI, corresponding to a fraction of their respective concentrations at lethal concentration.

**Morphological studies.** After treating mid log phase *C albicans* RSY150 with EOCs at MIC and 1/2 MIC for 4 h, cells were collected, washed with 0.01 M PBS (pH 7.4), stained with 0.01 $\mu$g/mL calcofluor white in 10% NaOH/water (81), and incubated for 5 min at room temperature. EOC-treated or untreated *C. albicans* were pipetted (5 $\mu$L) onto glass microscopic slides, covered with a clean coverslip, sealed with nail polish, and imaged on an Axio Observer Z1 inverted epifluorescence microscope (Oberkochen, Germany) at 63$\times$ magnification ($\lambda_{ex}$ = 365 nm; $\lambda_{em}$ = 435 nm). All fluorescence, including that for calcofluor white, was quantified using region-of-interest measurements in ZEN Blue Lite 2.3 image processing software, and the mean fluorescence intensities (in arbitrary units) were calculated from at least 300 individual cells per biological replicate.

**Membrane depolarization assay.** Changes in cytoplasmic membrane potential were measured using the membrane potential-sensitive probe, 3,3'-diethylthiadicarbocyanineiodide [DisC2(3)], as described previously (85, 86). Briefly, *C. albicans* RSY150 cultures ($\sim$10$^7$ CFU/mL) were exposed to eugenol or citral (MIC to 1/16 MIC), or the two at their FICI and 1/2 FICI in YPD media in a 96-well plate. DisC2 (3) dissolved in DMSO was added 5 min before the addition of test compounds to a final concentration of 2 $\mu$M, with the final concentration of DMSO limited to 1%. Amp B was used as a positive control. Changes in fluorescence intensity were measured before and 1 h after incubation at 30°C using a BioTek Synergy HTX multimode microplate reader (Winooski, VT) equipped with 560 nm excitation and 580 nm emission filters, and further examined by LSCM (LSCM 780; Carl Zeiss Oberkochen, Germany).

**Vacuole segregation and membrane integrity assay.** Vacuolar defects were assessed using the method described previously (41). Briefly, *C. albicans* RSY150 overnight cultures were suspended in freshly prepared YPD medium to a cell density of $\sim$10$^7$ CFU/mL, treated with EOCs at MIC and 1/2 MIC, or the two at their FICI for 4 h. Samples were then washed three times with PBS and transferred (5 $\mu$L) onto glass slides, sealed (nail polish) with clean coverslips, and imaged (Axio Observer Z1 inverted epifluorescence microscope; Oberkochen, Germany) using a 63$\times$ objective in bright field. At least 10 different fields of view were imaged for each biological replicate, of which 300 individual cells were counted and plotted. *C. albicans* strains incubated with only media and Amp B at MIC were used as negative and positive controls, respectively.

To further investigate the impact of EOCs on *C. albicans* vacuole membrane integrity, the aforementioned protocol was slightly modified with the addition of the lipophilic yeast vacuole membrane-specific dye MDY-64 (Invitrogen Y7536; Thermo Fisher Scientific) according to the manufacturer's procedure. Briefly, after the 4-h EOC exposure in 24-well microplates, treated and control cells were washed three times with PBS and resuspended in 200 $\mu$L of PBS, and MDY-64 dye was added to a final concentration of 10 $\mu$M. Samples were incubated for 3 min at room temperature in the dark, and microscopic slides were prepared and imaged by epifluorescence ($\lambda_{ex}$ = 451 nm; $\lambda_{em}$ = 497 nm) using a 63$\times$ objective. The results were expressed as the percentage of cells with intact, partially ruptured, and ruptured vacuole membranes, manually calculated from images captured in 10 different fields of view for a total of 100 cells each from three biological replicates.

**Mitochondrial membrane potential assay.** The impact of EOCs on mitochondria was determined using the membrane potential-sensitive probe MitoTracker Deep Red FM (Invitrogen, catalog no. M22426) according to published protocols (87). Briefly, *C. albicans* RBY1132 (parent strain of RSY150) were treated with EOCs for 4 h, harvested, washed, and resuspended in PBS to 10$^5$ cells/mL, from which 100 $\mu$L was transferred to 96 wells of a flat-bottom microplate. Plates were incubated with MitoTracker dye (5 $\mu$L, to 100 nM) at 30°C for 30 min in the dark, and stained cells were washed with PBS and viewed

using a Zeiss 780 LSCM ($\lambda_{ex}$ = 644 nm; $\lambda_{em}$ = 665 nm). *Candida* fluorescence intensities were quantified from 100 cells per biological replicate and plotted.

**Intracellular ROS assay.** *C. albicans* RBY1132 was assessed using the ROS-sensitive probe 2,7-dichlorodihydrofluroscein diacetate (DCFDA) according to a published protocol (88). Briefly, cell density was adjusted to $1 \times 10^7$ CFU/mL in YPD medium and exposed to EOCs (MIC to 1/16 MIC) or the positive control, 25 mM $H_2O_2$ at 30°C for 4 h. After a 30-min incubation with DCFDA (10 $\mu$mol/L), the cells were harvested, washed twice with PBS, and resuspended in the same buffer to $10^5$ cells/mL, and 100 $\mu$L of each suspension then transferred to a flat-bottom 96-well microplate. The fluorescence intensity ($\lambda_{ex}$ = 485 nm; $\lambda_{em}$ = 528 nm), directly indicating ROS levels, was measured in a BioTek Synergy HTX multimode microplate reader (at gain 35), and the cells treated at MIC, 1/2 MIC, and 1/4 MIC were transferred to microscopic slides for imaging and analysis by epifluorescence microscopy ($\lambda_{ex}$ = 485 nm; $\lambda_{em}$ = 528 nm).

**Cell cycle and microtubule analysis.** EOC-mediated cell cycle arrest and microtubule dysfunction were visualized in *C. albicans* RSY150 cells expressing Tub2-GFP ($\beta$-tubulin-tagged green fluorescent protein) and Htb-RFP (histone protein B-tagged red fluorescent protein), as previously described (41, 64). Briefly, a mid-logarithmic-phase culture with a cell density of $\sim$$10^7$ CFU/mL was exposed to the EOCs alone (MIC to 1/4 MIC) and in combination (FICI to 1/4 FICI) for 4 h. Treated and control cells were washed three times with PBS, transferred to glass slides sealed with clean coverslips, and imaged with a 63$\times$ objective on the Zeiss 780 LSCM using an argon laser ($\lambda_{ex}$ = 488 nm; $\lambda_{em}$ = 512 nm) for Tub2-GFP and HeNe laser ($\lambda_{ex}$ = 543 nm; $\lambda_{em}$ = 605 nm) for Htb-RFP. *C. albicans* incubated with media only or nocodazole were used as negative and positive controls, respectively. Cells were identified on the basis of nuclear organization (Htb-RFP) and enumerated in each cell cycle phase, while microtubules having varied degrees of Tub2-GFP incorporation (short, long) and diffuse tubulin were enumerated from the green fluorescence (75) of 250 cells. Fluorescence intensities and supermolecular MT lengths were calculated in ImageJ (http://rsb.info.nih.gov/ij/) from 100 cells per biological replicate.

**Membrane integrity and cellular leakage assays.** The impact of EOCs on membrane integrity and leakage of cellular content from *C. albicans* RSY150 was assessed as described in a previous report (64). Briefly, cells at mid-logarithmic phase were washed three times, resuspended to $\sim$$10^7$ CFU/mL in PBS, transferred to a 24-well plate containing EOCs (MIC to 1/16 MIC), and incubated at 30°C for 6 h with continuous shaking (200 rpm). Cells with no treatment served as controls and untreated cells in PBS and those exposed to Amp B served as a negative and positive controls, respectively. After incubation, the supernatant was diluted 1:10 with PBS and filtered (0.22 $\mu$m), and the leakage of cellular materials was assessed against the appropriate blank (EOCs in PBS) from the absorbance at 260 nm ($A_{260}$) using a Cary 100 BIO UV-VIS spectrophotometer (Varian, Midland, ON, Canada).

AFM (JPK Instruments, Berlin, Germany) was used to image the *C. albicans* cell surface and overall morphology at ultrahigh resolution in response to EOC exposure. Cells exposed to EOCs and Amp B (MIC to 1/4 MIC) and the combined EOCs (FICI and 1/2 FICI) were mounted on ultraclean coverslips (89) and imaged in QI mode with silicon nitride cantilevers (HYDRA6R-200NG; Nanosensors, Neuchatel, Switzerland) having calibrated spring constants ranging from 0.03 to 0.062 N/m, using a 7-$\mu$m Z-length and 100 $\mu$m/s raster scan (128 $\times$ 128 pixels). Surface roughness and cell height (JPK software) were measured at the midpoint of the cell, and the average cell height was calculated for at least 20 different cells from two different samples, as previously described (64).

**Live/dead cell assay.** The impact of EOCs on *C. albicans* RBY1132 was assessed using previously described methods (90), with slight modification. Briefly, $\sim$$10^7$ CFU/mL *C. albicans* were incubated with EOCs (MIC, 1/2 MIC, FICI, and 1/2 FICI) for 4 h in 24-well microplates, washed three times with PBS, resuspended in 200 $\mu$L of PBS, and 2 $\mu$L of PI in PBS was then added to the sample to a final concentration of 1 $\mu$g/mL. Samples were incubated for 30 min at 30°C in the dark, microscopic slides were prepared and imaged by epifluorescence Axio Observer (Oberkochen, Germany; $\lambda_{ex}$ = 493 nm; $\lambda_{em}$ = 636 nm), and the average fluorescence intensity was quantified (ZEN software) from 100 cells per biological replicate.

**Germ tube inhibition assay.** The effect of EOCs on *C. albicans* RSY150 germ tube formation was assessed following established protocols (91), with minor modifications. Mid-logarithmic-phase cultures of *C. albicans* grown overnight in YPD medium were adjusted to $\sim$$1 \times 10^7$ CFU/mL in prewarmed YPD with 10% fetal bovine serum (FBS) and deposited into a 12-well plate with the appropriate amounts of EOCs to achieve MIC and 1/2 MIC. *C. albicans* strains incubated with serum only or Amp B at MIC served as negative and positive controls, respectively. After 4 h of incubation at 37°C with gentle shaking, all agents were removed by centrifugation for 5 min (5,000 $\times$ *g*) accompanied by three PBS washes, and a small aliquot (5 $\mu$L) of culture was examined. The Axio Observer Z1 inverted epifluorescence microscope (Oberkochen, Germany) simultaneously revealed microtubules ($\lambda_{ex}$ = 488 nm; $\lambda_{em}$ = 512 nm for Tub2-GFP) and germ tubes (bright field). Approximately 100 cells from each biological replicate were assessed for germ tube formation, counted, and plotted.

***C. albicans* preformed biofilm assay.** A 100-$\mu$L aliquot of the *C. albicans* RSY150 suspension in YPD with 10% FBS was inoculated into 96-well plates and allowed to adhere by incubation with gentle shaking (75 rpm) for 90 min at 37°C (92), and nonadherent cells were then removed by gentle aspiration. Prewarmed YPD with 10% FBS was added to each well to allow biofilm to develop for 24 h prior to treating with EOCs. Planktonic cells were removed through two rounds of washing (200 $\mu$L of PBS), and fresh medium (100 $\mu$L, YPD in 10% FBS) either without or with EOCs (MIC to 1/16 MIC) was added, followed by static incubation at 37°C for 24 h. Wells containing only *C. albicans* in YPD broth with 10% FBS and Amp B served as a negative and positive controls, respectively. To determine the fraction of metabolically active *Candida* in the biofilm, planktonic cells were first aspirated, followed by the addition of MTT [3-(4,5-dimethylthiazol-2-yl)-2,5-diphenyl tetrazolium bromide] solution (50 $\mu$L, 1 mg/mL working solution) to each well, and the plates incubated for 4 h at 37°C. The MTT solution was removed with gentle

aspiration, 100 $\mu$L of dimethyl sulfoxide added to dissolve the dark blue formazan crystals, and the plates were further incubated for 15 min at room temperature. The $A_{570}$ was determined using a microplate reader (BioTek Epoch), and the percent inhibition of biofilm was calculated according to the following equation as described by Jadhav et al. (93):

$$\% \text{ Inhibition } = 100 - \left[\left\{\frac{A_{570} \text{ EO(C)}}{A_{570} \text{ Control}}\right\} \times 100\right]$$

A *C. albicans* (500 $\mu$L) biofilm grown on 24-well tissue culture plates treated with EOCs, as described above, followed by incubation at 37°C for 24 h, was initially assessed visually. Planktonic cells were removed by gentle aspiration and imaged with a stereomicroscope (Nikon SMZ 1500; Nikon, Japan) using a 4× objective to visualize the presence or the absence of biofilm.

**Filamentation and invasion assays.** Filamentation and agar invasion was assessed according to published protocols (91, 92) by spotting aliquots (2 $\mu$L) of *C. albicans* RSY150 suspension at mid-logarithmic phase ($\sim$1 × 10$^7$ CFU/mL) onto 12-well plates containing solid spider media (1% nutrient broth, 1% mannitol, and 0.2% K$_2$HPO$_4$) agar, with or without various concentrations of EOCs (1/16 to 1/2 MIC) or the two at their 1/2 FICI followed by incubation at 37°C for 6 days. For EOC pretreatment assays, mid-logarithmic-phase *Candida* samples were first exposed to eugenol, citral (1/2 to 1/16 MIC), and both agents at 1/2 FICI and incubated for 4 h at 30°C; the cells were then washed three times with PBS and resuspended to $\sim$10$^7$ CFU/mL in YPD. In both cases, colony formation and filamented growth at colony edges were determined using a stereomicroscope (Nikon SMZ 1500) and photographed using a digital camera.

**Statistical analyses.** Results, expressed as means ± the standard errors of the mean (SEM), from at least three biological replicates, were statistically analyzed using Prism software (v6.0; GraphPad Software, Inc., La Jolla, CA). A two-tailed unpaired Student's *t* test with a Welch's correction at a 95% confidence interval was used to determine the similarity between two data sets. Multiple treatments (EOC concentration) were assessed using a one-way analysis of variance (ANOVA) with a Dunnett's multiple comparison post test to compare all treated groups to untreated controls. A *P* value of <0.05 was considered statistically significant. *r* values, indicating linearity, were calculated in Excel using the built-in Pearson correlation coefficient.

## SUPPLEMENTAL MATERIAL

Supplemental material is available online only.
**SUPPLEMENTAL FILE 1**, PDF file, 0.8 MB.

## ACKNOWLEDGMENTS

We thank Richard J. Bennett (Department of Molecular Microbiology and Immunology, Brown University, Providence, RI) for the kind gift of the strains used in this study. This research was conducted on the traditional territories of the Nêhiyawak, Anihsinapek, Nakoda, Dakota, and Lakota peoples and the homeland of the Métis/Michif Nation.

This study was supported by Natural Science and Engineering Research Council (NSERC; 06649-2018) and Canada Foundation for Innovation grants to T.E.S. Z.S., and E.N. were partially supported by the Faculty of Graduate Studies and Research at the University of Regina. O.P. was supported by a NSERC undergraduate student research award, and T.S. was partially supported by the Faculty of Science.

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
