## [Reviewer comments · Microbiology Spectrum]

Microbiology Spectrum

Candida albicans ROS-dependent lethality and ROS-independent hyphal and biofilm inhibition by eugenol and citral

Zinnat Shahina, Easter Ndlovu, Omkaar Persaud, Taranum Sultana, and Tanya Dahms

Corresponding Author(s): Tanya Dahms, University of Regina

Review Timeline:

Submission Date:	August 13, 2022
Editorial Decision:	September 13, 2022
Revision Received:	October 14, 2022
Accepted:	October 17, 2022

Editor: Damian Krysan

Reviewer(s): The reviewers have opted to remain anonymous.

Transaction Report:

DOI: <https://doi.org/10.1128/spectrum.03183-22>

September 13, 2022

Dr. Tanya Elizabeth Susan Dahms
University of Regina
Chemistry and Biochemistry
3737 Wascana Parkway
Regina, SK S4S 0A2
Canada

Re: Spectrum03183-22 (*Candida albicans* ROS-dependent lethality and ROS-independent virulence inhibition by eugenol and citral)

Dear Dr. Tanya Elizabeth Susan Dahms:

The reviewers were somewhat split on their assessment. From the comments of reviewer 1, I would focus on FM-464 if you want to maintain the finding of vacuole effects. I would also discuss the relevance of the findings of effects at MIC. One easy way to address this is to do PI staining of the cells at MIC at the time point in which the phenotypic assay is performed or do a time course to assess the kinetics of killing. I also agree that the mention of virulence in the title is not supported.

Link Not Available

Sincerely,

Damian Krysan

Journals Department
Reviewer comments:

Reviewer #1 (Comments for the Author):

The manuscript by Shahina et al characterizes the response of the fungal pathogen *Candida albicans* to two different essential oils, eugenol and citral, both alone and in combination. The authors conclude that very high concentrations of these compounds results in membrane damage, oxidative stress, vacuole segregation, microtubule dysfunction, and cell cycle arrest. Prior to

publication in *Microbiology Spectrum*, several points should be addressed:

Major points

- 1) Many of the statements made throughout the manuscript are overreaching. For example, the last sentence of the abstract (lines 23-24) states, "This study links the inhibition of virulence factors with vacuole and microtubule dysfunction." The authors should revise the text so that their conclusions are accurate. This sentence should state, "High concentrations of eugenol and citral lead to vacuole and microtubule dysfunction."
- 2) The title of the manuscript needs to be changed to accurately reflect the data. Both citral and eugenol were shown to cause cell death however only eugenol resulted in ROS accumulation. Therefore, having the title describe "ROS-dependent lethality" is not substantiated. Further, "virulence" wasn't formally assessed, only hyphal formation and biofilm formation in vitro.
- 3) Figure 1: The authors conclude the combination of eugenol and citral has an additive effect based on an FICI of 0.9 ± 0.14 and the isobologram shown in Fig. 1c. Based on the error it is unclear as to whether there is any interaction between the two compounds at all. It is also unclear what the error represents. Is this a deviation from biological replicates and if so how many? Further, for the isobologram shown in Figure 1c, no conclusion can be made regarding the shape of the curve as the axis need to be formatted correctly. No data points for individual drugs are included and the axis are not symmetrical (having the increments on the y-axis vary by 100 $\mu\text{g}/\text{mL}$ with a shorter axis while the increments on the x-axis vary by 50 $\mu\text{g}/\text{mL}$ and be longer skews the shape of the curve and therefore the interpretation). It would be more helpful to show the growth data in a heat map format for all drug combinations tested. In addition, details as to how many replicates were performed for this experiment are needed.
- 4) For Figure 2, the authors conclude chitin content was elevated for eugenol and AmpB. Including a more suitable positive control (a cell wall perturbing agent such as caspofungin or calcofluor white) would be helpful as any increase was marginal and was not clear based on the microscopy images.
- 5) For Figure 3c, the authors conclude treatment with AmpB, eugenol, and citral lead to increase in vacuole segregation based on bright field microscopy. It is unclear how the authors know that a "segmented vacuole" is what is observed/indicated with the arrow. It would be beneficial to use a membrane dye such as FM-464 to stain all cellular membranes to visualize the effects of the compounds more accurately on intracellular membrane organelles.
- 6) For Figure 5, the authors show that treatment with eugenol and citral and MIC concentrations leads to abnormal microtubules. However, is this simply a generic response to growth inhibition/cell death? Would treatment with any compound at its MIC generate the same effect? Similarly, would treatment with any compound at its MIC result in cell cycle arrest as is quantified in Figure 5e?
- 7) For Figure 7, is the effect of eugenol and citral on hyphal formation simply because the cells aren't viable? Based on the microscopy, images shown at 1/2 MIC or 1/2 FICI look comparable to the control with the only block in filamentation observed at MIC.

Minor points

- 1) Line 29: "fungi" should be "fungus"
- 2) Line 43: "organ transplants" should be changed to "organ transplant recipients"
- 3) As much as possible, do not use abbreviations. As an example, "FOV" is not a standard abbreviation that will be understood by all readers.
- 4) Line 438, "0.1 as additive" should read "1.0 as additive."

Reviewer #2 (Comments for the Author):

In this paper, the authors investigated the antifungal capabilities of the essential oil compounds (EOCs) eugenol and citral. They showed that these compounds had an additive effect in inhibiting the growth of the fungal pathogen *C. albicans*. They show altered growth in response to both EOCs as an increase in pseudohyphae formation and decreases in mycelia and biofilm formation. The authors also show morphological defects such as altered vacuolar segregation, mitochondrial depolarization, altered membrane integrity, and microtubule dysfunction resulting in cell cycle arrest. The authors conclude that the combination of both eugenol and citral act in a ROS-independent manner to inhibit virulence factors primarily through microtubule disruption. Overall the conclusion of this paper is supported by the data presented. However, there are a significant number of run-on sentences throughout the text that make interpreting the findings of this paper difficult. Breaking up these sentences would aid both clarity and strengthen the conclusions of the paper.

- Line 100: The comma separating "regular yeast morphology" and "with budding yeast cells" is not necessary
- Figure 2 panel D: The Y axis should refer to "% pseudohyphae forming cells" instead of chain forming to be consistent with what is said in the text (Line 102-103).
- Figure 3d and Line 132: It may be more accurate to say that vacuolar segregation in response to both EOCs and FICI was like eugenol at MIC as opposed to being mainly contributed to eugenol, as the FICI is composed of only 1/2 the concentration eugenol at MIC. Thus, synergy between both the citral and eugenol must be driving the increase in vacuolar segregation
- Line 133-136: The findings presented in this section would be clearer if broken up into individual sentences.
- Line 183: "..., indicating disruption of..." This should be its own sentence to as to state the meaning of the findings in the sentence before it.
- Line 193: "RSY150 control cells showed..." run-on sentence, separating this section into multiple sentence would enhance the clarity and importance of the findings presented

- Figure 5B and C: having the 1/4 and 1/2 next to FICI is redundant
- Line 239: A sentence describing what the PI fluorescence assay is assessing would be beneficial.
- 241 to 242: It is claimed that PI uptake was significant for citral at MIC and for both EOCs at FICI and that uptake at 1/2 MIC was poor. However, PI fluorescence at 1/2 MIC appears to be significant by the statistical assessment performed.

Staff Comments:

Preparing Revision Guidelines

Please return the manuscript within 60 days; if you cannot complete the modification within this time period, please contact me. If you do not wish to modify the manuscript and prefer to submit it to another journal, please notify me of your decision immediately so that the manuscript may be formally withdrawn from consideration by Microbiology Spectrum.

Reviewer #1 (Comments for the Author):

The manuscript by Shahina et al characterizes the response of the fungal pathogen *Candida albicans* to two different essential oils, eugenol and citral, both alone and in combination. The authors conclude that very high concentrations of these compounds results in membrane damage, oxidative stress, vacuole segregation, microtubule dysfunction, and cell cycle arrest. Prior to publication in *Microbiology Spectrum*, several points should be addressed:

Major points

1) Many of the statements made throughout the manuscript are overreaching. For example, the last sentence of the abstract (lines 23-24) states, "This study links the inhibition of virulence factors with vacuole and microtubule dysfunction." The authors should revise the text so that their conclusions are accurate. This sentence should state, "High concentrations of eugenol and citral lead to vacuole and microtubule dysfunction."

Thank you for pointing out this error, which was the result of combining several sentences through multiple edits, ultimately skewing the meaning of the sentence, which we have corrected to read: "This study shows that eugenol and citral can induce vacuole and microtubule dysfunction, along with the inhibition of hyphal and biofilm formation." (lines 24-25)

2) The title of the manuscript needs to be changed to accurately reflect the data. Both citral and eugenol were shown to cause cell death however only eugenol resulted in ROS accumulation. Therefore, having the title describe "ROS-dependent lethality" is not substantiated. Further, "virulence" wasn't formally assessed, only hyphal formation and biofilm formation in vitro.

Thank you for these helpful suggestions. The majority of this study examined 4 h EOC exposure, which indeed does not cause ROS accumulation for citral, but exposure to citral for 24 h (typical time point for a MIC assay) did induce ROS formation (see supplementary FIG.S2), and we now describe this in the text (lines 188-189).

We have corrected the title: "*Candida albicans* ROS-dependent lethality and ROS-independent hyphal and biofilm inhibition by eugenol and citral", which we think captures the data. (line 2)

3) Figure 1: The authors conclude the combination of eugenol and citral has an additive effect based on an FICI of 0.9 +/- 0.14 and the isobologram shown in Fig. 1c. Based on the error it is unclear as to whether there is any interaction between the two compounds at all. It is also unclear what the error represents. **Is this a deviation from biological replicates and if so how many?** Further, for the isobologram shown in Figure 1c, **no conclusion can be made regarding the shape of the curve as the axis need to be formatted correctly.** No data points for individual drugs are included and the axis are not symmetrical (having the increments on the y-axis vary by 100 ug/mL with a shorter axis while the increments on the x-axis vary by 50 ug/mL and be longer skews the shaper of the curve and therefore the interpretation). It would be more helpful to show the growth data in a heat map format for all drug combinations tested. In addition, details as to how many replicates were performed for this experiment are needed.

Thank you for catching this - looking back at the data, we realize that the FICI had been reported incorrectly. The checker board assay was assessed from three biological replicates, giving FICI values of 0.66, 0.90 and 0.90, so in fact the mean is 0.83, with a SEM of 0.14. We have corrected these values in both the abstract (line 18) and results (line 90) sections. In the statistical analysis section we had indicated “Results expressed as the mean \pm standard error of the mean (SEM), from at least three independent experiments,” but to further clarify, we have changed this to read “three biological replicates”.

Typically in the literature, isobolograms are reported as scatter plots with skewed axes (left image below), and since we had incorrectly curve fitted the data the line was curved so we have corrected this. Certainly an ideal isobologram is corrected for the MIC values (right image below), giving rise to symmetrical axes. We have changed the graph accordingly in this figure, but would argue that both (below) show the exact same pattern and are equally interpretable. The non-linearity on the concave side, shows the additive effect.

Our supplementary data includes FIG S2 entitled “Checkerboard microtiter plate assay shows the impact of eugenol and citral on *C. albicans* RSY150”, and although we uploaded this to the journal website, we are not sure if the reviewers had access to that information.

4) For Figure 2, the authors conclude **chitin content was elevated for eugenol and AmpB. Including a more suitable positive control (a cell wall perturbing agent such as caspofungin or calcofluor white)** would be helpful as any increase was marginal and was not clear based on the microscopy images.

CFW is in fact calcofluor white, which relates to your comment about abbreviations, so we have changed that to read “calcofluor white” in the figure caption (Line 124) and elsewhere. We argue that compared to control, higher fluorescence intensity is visible in the representative images for eugenol and AmpB. Further, the fluorescence intensity data shown in Figure 2b was quantified from three biological replicates, with 300 cells per replicate and statistically evaluated using an unpaired Student’s *t*-test, as specified in the figure caption (lines 129-131).

5) For Figure 3c, the authors conclude treatment with AmpB, eugenol, and citral lead to increase in vacuole segregation based on bright field microscopy. It is unclear how the authors know that a "segmented vacuole" is what is observed/indicated with the arrow. It would be beneficial to use a membrane dye such as **FM-464** to stain all cellular membranes to visualize the effects of the compounds more accurately on intracellular membrane organelles.

In the manuscript, we do not use the word segmented, rather segregated, meaning divided. In the control images we can see large intact vacuoles, which we now explicitly state in the results section (lines 121, 132), while we think that small segregated vacuoles are clearly visible in the treated cells.

We thank the reviewer for the suggestion of conducting experiments with a membrane dye. We have accordingly incorporated data from new assays using the vacuole membrane specific dye, MDY64 (Fig 4a), described in the results (Figure 4, lines 169-180) and methods (lines 522-533) sections.

6) For Figure 5, the authors show that treatment with eugenol and citral and MIC concentrations leads to abnormal microtubules. However, is this simply a generic response to growth inhibition/cell death? **Would treatment with any compound at its MIC generate the same effect? Similarly, would treatment with any compound at its MIC result in cell cycle arrest as is quantified in Figure 5e?**

Thank you for pointing this out, and we have added additional text in relation to tubulin and microtubules (lines 216-219). We do not believe this to be simply a generic response indicative of cell death since we also see microtubule disruption that is ROS-independent at sub-lethal exposure. We have included two additional references that document microtubule disruption with exposure to other essential oils and their components (line 399). Although treatment with any compound that induces cell death may also result in microtubule disruption, here we see microtubule disruption under conditions that do not induce cell death. It could be that any compound at its MIC may also result in cell cycle arrest, but the interesting observation is at sub-MIC levels that are ROS-independent.

7) For Figure 7, is the effect of eugenol and citral on hyphal formation simply because the cells aren't viable? **Based on the microscopy, images shown at 1/2 MIC or 1/2 FICI look comparable to the control with the only block in filamentation observed at MIC.**

In this figure (now Figure 8), we would argue that the cells are viable because they continually formed pseudohyphae. Please note that the control images in Figure 7a show typical germ tubes with healthy microtubules (green), while exposure at 1/2 MIC and 1/2 FICI show pseudohyphae with the majority having disperse tubulin (green). Certainly there are some microtubule clusters in the pseudohyphae at 1/2 FICI, but with an equal amount that are disperse and no sign of elongated microtubules, as viewed in controls.

Minor points

1) **Line 29:** "fungi" should be "fungus"

Thank you for catching this oversight. We have changed it accordingly.

2) **Line 43:** "organ transplants" should be changed to "organ transplant recipients"

Thank you, we have corrected this.

3) As much as possible, do not use abbreviations. As an example, "FOV" is not a standard abbreviation that will be understood by all readers.

We have tried to remove extraneous abbreviations.

4) **Line 438**, "0.1 as additive" should read "1.0 as additive.

Thank you for catching this error. This has been corrected and now appears on line 478.

Reviewer #2 (Comments for the Author):

In this paper, the authors investigated the antifungal capabilities of the essential oil compounds (EOCs) eugenol and citral. They showed that these compounds had an additive effect in inhibiting the growth of the fungal pathogen *C. albicans*. They show altered growth in response to both EOCs as an increase in pseudohyphae formation and decreases in mycelia and biofilm formation. The authors also show morphological defects such as altered vacuolar segregation, mitochondrial depolarization, altered membrane integrity, and microtubule dysfunction resulting in cell cycle arrest. The authors conclude that the combination of both eugenol and citral act in a ROS-independent manner to inhibit virulence factors primarily through microtubule disruption. Overall the conclusion of this paper is supported by the data presented. However, there are a significant number of run-on sentences throughout the text that make interpreting the findings of this paper difficult. Breaking up these sentences would aide both clarity and strengthen the conclusions of the paper.

We appreciate this suggestion and have tried to make each sentence more succinct. We have split many into two separate sentences, or heavily edited sentences for concision.

- **Line 100**: The comma separating "regular yeast morphology" and "with budding yeast cells" is not necessary

Thank you for catching this. We have removed the comma (line 102).

- Figure 2 panel D: The Y axis should refer to "**% pseudohyphae forming cells**" instead of chain forming to be consistent with what is said in the text (Line 102-103).

We thank you for catching this inconsistency. We have changed the labels on Figure 2 so that they accurately read as "% unipolar pseudohyphae" (clusters) and "% chain-forming pseudohyphae", as the bar graphs in Figure 2 c,d quantify these distinct pseudohyphal morphologies.

- Figure 3d and Line 132: It may be more accurate to say that vacuolar segregation in response to both EOCs and FICI was like eugenol at MIC as opposed to being mainly contributed to eugenol, as the FICI is composed of only 1/2 the concentration eugenol at MIC. Thus, **synergy between** both the citral and eugenol must be driving the increase in vacuolar segregation.

We thank you for this suggestion, which we have incorporated (lines 135-138).

- **Line 133-136**: The findings presented in this section would be clearer if broken up into individual sentences.

We have corrected this accordingly (lines 121, 132, 135-141,144-145).

- **Line 183**: ..., indicating disruption of..." This should be its own sentence to as to state the meaning of the findings in the sentence before it.

We thank you for this suggestion, and we have modified accordingly (lines 216-219).

- **Line 193:** "RSY150 control cells showed..." run-on sentence, separating this section into multiple sentence would enhance the clarity and importance of the findings presented
Although we kept this as one sentence, it has been edited for concision (lines 228-230).

- **Figure 5B and C:** having the 1/4 and 1/2 next to FICI is redundant
We not only determined the biological impact of exposure to the two components at their FICI, but we also examined the impacts at 1/2 and 1/4 FICI, the component concentrations at 1/4 and 1/2 their respective concentrations of their combined lethal concentration. Thus, we need to include 1/4 and 1/2 for comparison with impacts at FICI. We have clarified this in the methods section (Lines 485-486)

- **Line 239:** A sentence describing what the PI fluorescence assay is assessing would be beneficial.
Thank you for pointing out this omission, which we have now described (lines 275-277).

- **241 to 242:** It is claimed that PI uptake was significant for citral at MIC and for both EOCs at FICI and that uptake at 1/2 MIC was poor. However, PI fluorescence at 1/2 MIC appears to **be significant** by the statistical assessment performed.
You are correct. We have reviewed the data and corrected the text accordingly (Lines 280-281).

October 17, 2022

Dr. Tanya Elizabeth Susan Dahms
University of Regina
Chemistry and Biochemistry
3737 Wascana Parkway
Regina, SK S4S 0A2
Canada

Re: Spectrum03183-22R1 (Candida albicans ROS-dependent lethality and ROS-independent hyphal and biofilm inhibition by eugenol and citral)

Dear Dr. Tanya Elizabeth Susan Dahms:

Thank you for your responsive revision and carefully addressing the points and suggestions raised by the reviewers.

Your manuscript has been accepted, and I am forwarding it to the ASM Journals Department for publication. You will be notified when your proofs are ready to be viewed.

Sincerely,

Damian Krysan
Editor, Microbiology Spectrum
